# GILBO: One Metric to Measure Them All

**Alexander A. Alemi,**[*] **Ian Fischer**[*]
Google AI
{alemi,iansf}@google.com

## Abstract

We propose a simple, tractable lower bound on the mutual information contained in the joint generative density of any latent variable generative model: the GILBO (*Generative Information Lower BOund*). It offers a data-independent measure of the complexity of the learned latent variable description, giving the log of the effective description length. It is well-defined for both VAEs and GANs. We compute the GILBO for 800 GANs and VAEs each trained on four datasets (MNIST, FashionMNIST, CIFAR-10 and CelebA) and discuss the results.

## 1   Introduction

GANs (Goodfellow et al., 2014) and VAEs (Kingma & Welling, 2014) are the most popular latent variable generative models because of their relative ease of training and high expressivity. However *quantitative* comparisons across different algorithms and architectures remains a challenge. VAEs are generally measured using the ELBO, which measures their fit to data. Many metrics have been proposed for GANs, including the INCEPTION score (Gao et al., 2017), the FID score (Heusel et al., 2017), independent Wasserstein critics (Danihelka et al., 2017), birthday paradox testing (Arora & Zhang, 2017), and using Annealed Importance Sampling to evaluate log-likelihoods (Wu et al., 2017), among others.

Instead of focusing on metrics tied to the data distribution, we believe a useful additional independent metric worth consideration is the complexity of the trained generative model. Such a metric would help answer questions related to overfitting and memorization, and may also correlate well with sample quality. To work with both GANs and VAEs our metric should not require a tractable joint density $p(x, z)$. To address these desiderata, we propose the GILBO.

## 2   GILBO: Generative Information Lower BOund

A symmetric, non-negative, reparameterization independent measure of the information shared between two random variables is given by the mutual information:

$$I(X; Z) = \iint dx\, dz\, p(x, z) \log \frac{p(x, z)}{p(x)p(z)} = \int dz\, p(z) \int dx\, p(x|z) \log \frac{p(z|x)}{p(z)} \geq 0. \quad (1)$$

$I(X; Z)$ measures how much information (in nats) is learned about one variable given the other. As such it is a measure of the complexity of the generative model. It can be interpreted (when converted to bits) as the reduction in the number of yes-no questions needed to guess $X = x$ if you observe $Z = z$ and know $p(x)$, or vice-versa. It gives the log of the *effective description length* of the generative model. This is roughly the log of the number of distinct sample pairs (Tishby & Zaslavsky, 2015). $I(X; Z)$ is well-defined even for continuous distributions. This contrasts with the continuous entropy $H(X)$ of the marginal distribution, which is not reparameterization independent (Marsh,

---

[*]Authors contributed equally.

2013). $I(X; Z)$ is intractable due to the presence of $p(x) = \int dz\, p(z)p(x|z)$, but we can derive a tractable variational lower bound (Agakov, 2006):

$$I(X; Z) = \iint dx\, dz\, p(x, z) \log \frac{p(x, z)}{p(x)p(z)} \tag{2}$$

$$= \iint dx\, dz\, p(x, z) \log \frac{p(z|x)}{p(z)} \tag{3}$$

$$\geq \iint dx\, dz\, p(x, z) \log p(z|x) - \int dz\, p(z) \log p(z) - \mathrm{KL}[p(z|x)||e(z|x)] \tag{4}$$

$$= \int dz\, p(z) \int dx\, p(x|z) \log \frac{e(z|x)}{p(z)} = \mathbb{E}_{p(x,z)} \left[ \log \frac{e(z|x)}{p(z)} \right] \equiv \text{GILBO} \leq I(X; Z) \tag{5}$$

We call this bound the GILBO for *Generative Information Lower BOund*. It requires learning a tractable variational approximation to the intractable posterior $p(z|x) = p(x, z)/p(x)$, termed $e(z|x)$ since it acts as an *encoder* mapping from data to a prediction of its associated latent variables.[2] As a variational approximation, $e(z|x)$ depends on some parameters, $\theta$, which we elide in the notation.

The encoder $e(z|x)$ performs a regression for the inverse of the GAN or VAE generative model, approximating the latents that gave rise to an observed sample. This encoder should be a tractable distribution, and must respect the domain of the latent variables, but does not need to be reparameterizable as no sampling from $e(z|x)$ is needed during training. We suggest the use of $(-1, 1)$ remapped Beta distributions in the case of uniform latents, and Gaussians in the case of Gaussian latents. In either case, training the variational encoder consists of simply generating pairs of $(x, z)$ from the trained generative model and maximizing the likelihood of the encoder to generate the observed $z$, conditioned on its paired $x$, divided by the likelihood of the observed $z$ under the generative model's prior, $p(z)$. For the GANs in this study, the prior was a fixed uniform distribution, so the $\log p(z)$ term contributes a constant offset to the variational encoder's likelihood. Optimizing the GILBO for the parameters of the encoder gives a lower bound on the true generative mutual information in the GAN or VAE. Any failure to converge or for the approximate encoder to match the true distribution does not invalidate the bound, it simply makes the bound looser.

The GILBO contrasts with the *representational mutual information* of VAEs defined by the data and encoder, which motivates VAE objectives (Alemi et al., 2017). For VAEs, both lower and upper variational bounds can be defined on the representational joint distribution $(p(x)e(z|x))$. These have demonstrated their utility for cross-model comparisons. However, they require a tractable posterior, preventing their use with most GANs. The GILBO provides a theoretically-justified and dataset-independent metric that allows direct comparison of VAEs and GANs.

The GILBO is entirely independent of the *true* data, being purely a function of the generative joint distribution. This makes it distinct from other proposed metrics like estimated marginal log likelihoods (often reported for VAEs and very expensive to estimate for GANs) (Wu et al., 2017)[3], an independent Wasserstein critic (Danihelka et al., 2017), or the common INCEPTION (Gao et al., 2017) and FID (Heusel et al., 2017) scores which attempt to measure how well the generated samples match the observed true data samples. Being independent of data, the GILBO does not directly measure sample quality, but extreme values (either low or high) correlate with poor sample quality, as demonstrated in the experiments below.

Similarly, in Im et al. (2018), the authors propose using various GAN training objectives to quantitatively measure the performance of GANs on their own generated data. Interestingly, they find that evaluating GANs on the same metric they were trained on gives paradoxically weaker performance – an LS-GAN appears to perform worse than a Wasserstein GAN when evaluated with the least-squares metric, for example, even though the LS-GAN otherwise outperforms the WGAN. If this result holds in general, it would indicate that using the GILBO during training might result in less-interpretable evaluation GILBOs. We do not investigate this hypothesis here.

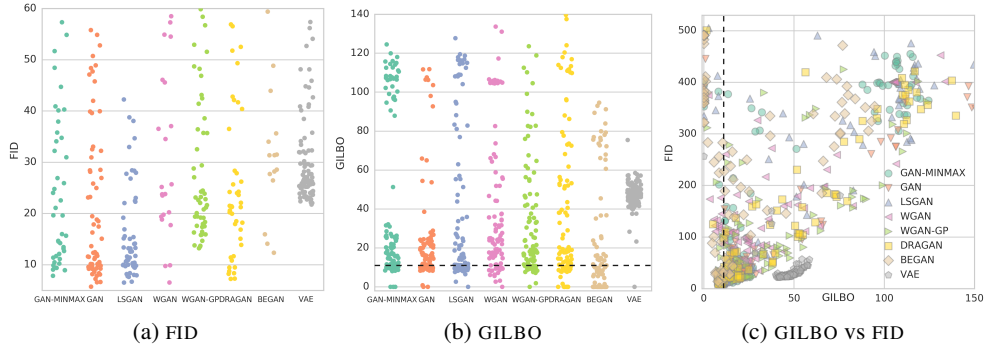

Figure 1: (a) Recreation of Figure 5 (left) from Lucic et al. (2017) showing the distribution of FID scores for each model on MNIST. Points are jittered to give a sense of density. (b) The distribution of GILBO scores. (c) FID vs GILBO.

Although the GILBO doesn't directly reference the dataset, the dataset provides useful signposts. First is at $\log C$, the number of distinguishable classes in the data. If the GILBO is lower than that, the model has almost certainly failed to learn a reasonable model of the data. Another is at $\log N$, the number of training points. A GILBO near this value may indicate that the model has largely memorized the training set, or that the model's capacity happens to be constrained near the size of the training set. At the other end is the entropy of the data itself ($H(X)$) taken either from a rough estimate, or from the best achieved data log likelihood of any known generative model on the data. Any reasonable generative model should have a GILBO no higher than this value.

Unlike other metrics, GILBO does not monotonically map to quality of the generated output. Both extremes indicate failures. A vanishing GILBO denotes a generative model with vanishing complexity, either due to independence of the latents and samples, or a collapse to a small number of possible outputs. A diverging GILBO suggests over-sensitivity to the latent variables.

In this work, we focus on variational approximations to the generative information. However, other means of estimating the GILBO are also valid. In Section 4.3 we explore a computationally-expensive method to find a very tight bound. Other possibilities exist as well, including the recently proposed *Mutual Information Neural Estimation* (Belghazi et al., 2018) and *Contrastive Predictive Coding* (Oord et al., 2018). We do not explore these possibilities here, but any valid estimator of the mutual information can be used for the same purpose.

## 3 Experiments

We computed the GILBO for each of the 700 GANs and 100 VAEs tested in Lucic et al. (2017) on the MNIST, FashionMNIST, CIFAR and CelebA datasets in their *wide range* hyperparameter search. This allowed us to compare FID scores and GILBO scores for a large set of different GAN objectives on the same architecture. For our encoder network, we duplicated the discriminator, but adjusted the final output to be a linear layer predicting the $64 \times 2 = 128$ parameters defining a $(-1, 1)$ remapped Beta distribution (or Gaussian in the case of the VAE) over the latent space. We used a Beta since all of the GANs were trained with a $(-1, 1)$ 64-dimensional uniform distribution. The parameters of the encoder were optimized for up to 500k steps with ADAM (Kingma & Ba, 2015) using a scheduled multiplicative learning rate decay. We used the same batch size (64) as in the original training. Training time for estimating GILBO is comparable to doing FID evaluations (a few minutes) on the small datasets (MNIST, FashionMNIST, CIFAR), or over 10 minutes for larger datasets and models (CelebA).

In Figure 1 we show the distributions of FID and GILBO scores for all 800 models as well as their scatter plot for MNIST. We can immediately see that each of the GAN objectives collapse to GILBO $\sim 0$ for some hyperparameter settings, but none of the VAEs do. In Figure 2 we show generated samples from all of the models, split into relevant regions. A GILBO near zero signals a failure of the model to make any use of its latent space (Figure 2a).

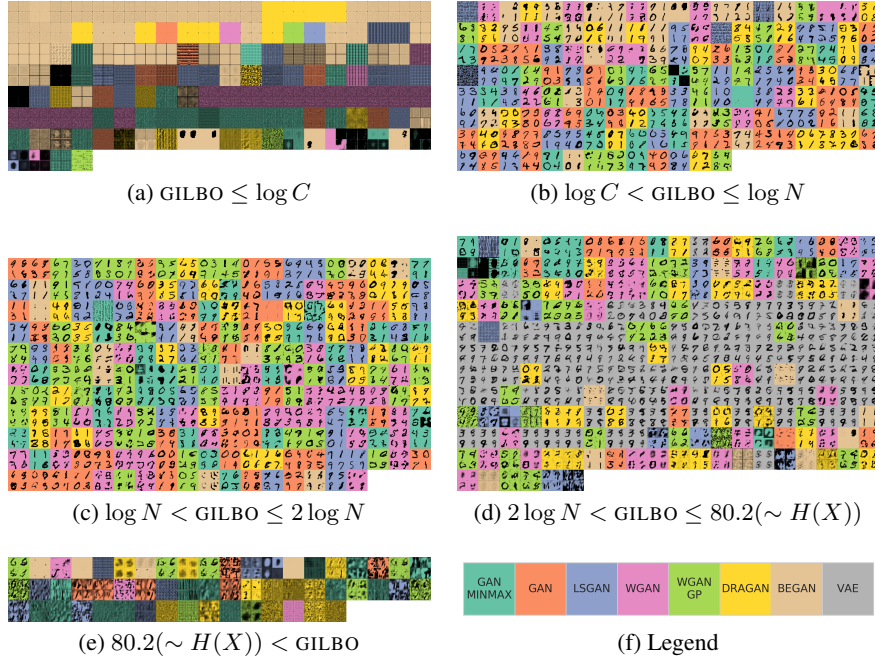

(a) GILBO $\leq \log C$

(b) $\log C <$ GILBO $\leq \log N$

(c) $\log N <$ GILBO $\leq 2 \log N$

(d) $2 \log N <$ GILBO $\leq 80.2(\sim H(X))$

(e) $80.2(\sim H(X)) <$ GILBO

(f) Legend

Figure 2: Samples from all models sorted by increasing GILBO in raster order and broken up into representative ranges. The background colors correspond to the model family (Figure 2f). Note that all of the VAE samples are in (d), indicating that the VAEs achieved a non-trivial amount of complexity. Also note that most of the GANs in (d) have poor sample quality, further underscoring the apparent difficulty these GANs have maintaining high visual quality without indications of training set memorization.

The best performing models by FID all sit at a GILBO $\sim 11$ nats. An MNIST model that simply memorized the training set and partitioned the latent space into 50,000 unique outputs would have a GILBO of $\log 50{,}000 = 10.8$ nats, so the cluster around 11 nats is suspicious. Since mutual information is invariant to any invertible transformation, a model that partitioned the latent space into 50,000 bins, associated each with a training point and then performed some random elastic transformation but with a magnitude low enough to not turn one training point into another would still have a generative mutual information of 10.8 nats. Larger elastic transformations that could confuse one training point for another would only act to lower the generative information. Among a large set of hyperparameters and across 7 different GAN objectives, we notice a conspicuous increase in FID score as GILBO moves away from $\sim 11$ nats to either side. This demonstrates the failure of these GANs to achieve a meaningful range of complexities while maintaining visual quality. Most striking is the distinct separation in GILBOs between GANs and VAEs. These GANs learn less complex joint densities than a vanilla VAE on MNIST at the same FID score.

Figures 3 to 5 show the same plots as in Figure 1 but for the FashionMNIST, CIFAR-10 and CelebA datasets respectively. The best performing models as measured by FID on FashionMNIST continue to have GILBOs near $\log N$. However, on the more complex CIFAR-10 and CelebA datasets we see nontrivial variation in the complexities of the trained GANs with competitive FID. On these more complex datasets, the visual performance (e.g. Figure 8) of the models leaves much to be desired. We speculate that the models' inability to acheive high visual quality is due to insufficient model capacity for the dataset.

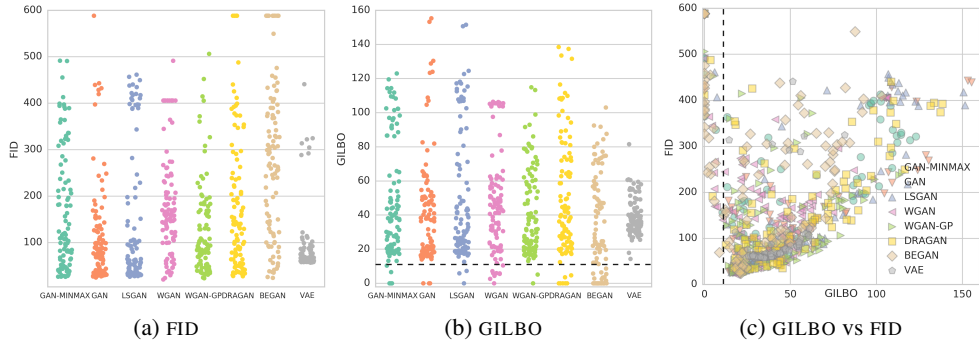

(a) FID            (b) GILBO            (c) GILBO vs FID

Figure 3: A recreation of Figure 1 for the Fashion MNIST dataset.

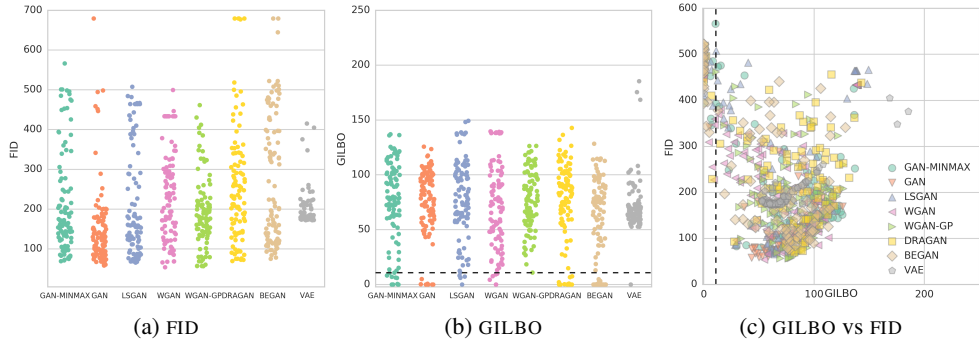

(a) FID            (b) GILBO            (c) GILBO vs FID

Figure 4: A recreation of Figure 1 for the CIFAR dataset.

# 4 Discussion

## 4.1 Reproducibility

While the GILBO is a valid lower bound regardless of the accuracy of the learned encoder, its utility as a metric naturally requires it to be comparable across models. The first worry is whether it is reproducible in its values. To address this, in Figure 6 we show the result of 128 different training runs to independently compute the GILBO for three models on CelebA. In each case the error in the measurement was below 2% of the mean GILBO and much smaller in variation than the variations between models, suggesting comparisons between models are valid if we use the same encoder architecture $(e(z|x))$ for each.

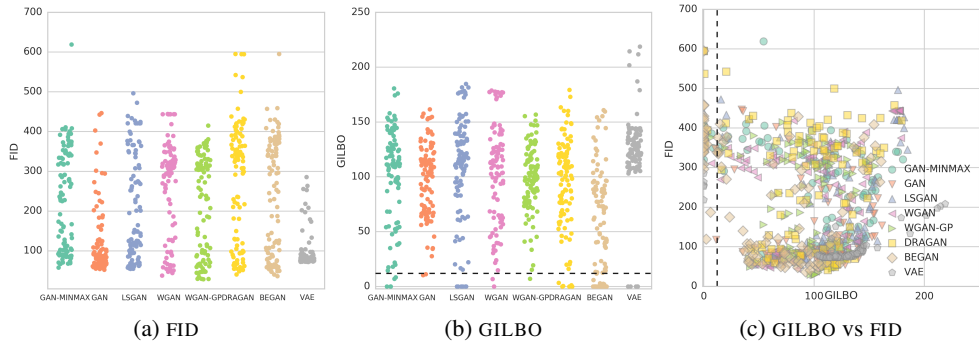

(a) FID            (b) GILBO            (c) GILBO vs FID

Figure 5: A recreation of Figure 1 for the CelebA dataset.

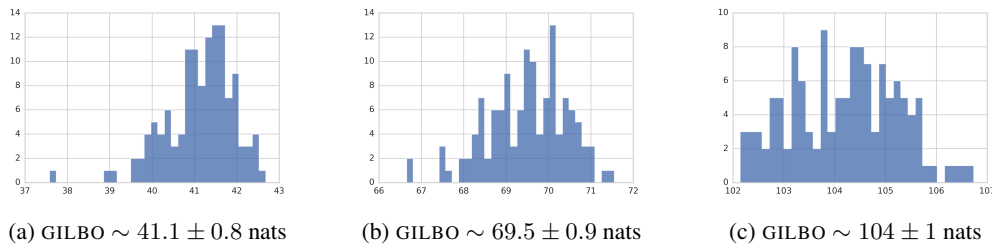

(a) GILBO $\sim 41.1 \pm 0.8$ nats     (b) GILBO $\sim 69.5 \pm 0.9$ nats     (c) GILBO $\sim 104 \pm 1$ nats

Figure 6: Measure of the reproducibility of the GILBO for the three models visualized in Figure 8. For each model we independently measured the GILBO 128 times.

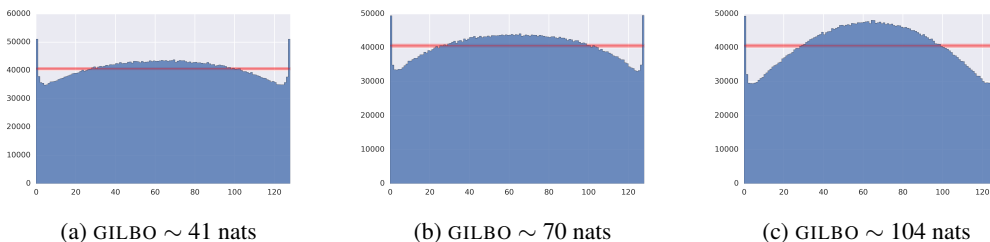

(a) GILBO $\sim 41$ nats     (b) GILBO $\sim 70$ nats     (c) GILBO $\sim 104$ nats

Figure 7: Simulation-based calibration (Talts et al., 2018) of the variational encoder for the same three models as in Figure 6. Shown are histograms of the ranking statistic for how many of 128 samples from the encoder are less than the true $z$ used to generate the figure, aggregated over the 64 dimensional latent vector for 1270 batches of 64 samples each. Shown in red is the 99% confidence interval for a uniform distribution, the expected result if $e(z|x)$ was the true $p(z|x)$. The systematic ∩-shape denotes overdispersion in the approximation.

## 4.2   Tightness

Another concern would be whether the learned variational encoder was a good match to the true posterior of the generative model ($e(z|x) \sim p(z|x)$). Perhaps the model with a measured GILBO of 41 nats simply had a harder to capture $p(z|x)$ than the GILBO $\sim 104$ nat model. Even if the values were reproducible between runs, maybe there is a systemic bias in the approximation that differs between different models.

To test this, we used the *Simulation-Based Calibration* (SBC) technique of Talts et al. (2018). If one were to implement a cycle, wherein a single draw from the prior $z' \sim p(z)$ is decoded into an image $x' \sim p(x|z')$ and then inverted back to its corresponding latent $z_i \sim p(z|x')$, the rank statistic $\sum_i \mathbb{I}[z_i < z']$ should be uniformly distributed. Replacing the true $p(z|x')$ with the approximate $e(z|x)$ gives a visual test for the accuracy of the approximation. Figure 7 shows a histogram of the rank statistic for 128 draws from $e(z|x)$ for each of 1270 batches of 64 elements each drawn from the 64 dimensional prior $p(z)$ for the same three GANs as in Figure 6. The red line denotes the 99% confidence interval for the corresponding uniform distribution. All three GANs show a systematic ∩-shaped distribution denoting overdispersion in $e(z|x)$ relative to the true $p(z|x)$. This is to be expected from a variational approximation, but importantly the degree of mismatch seems to correlate with the scores, not anticorrelate. It is likely that the 41 nat GILBO is a more accurate lower bound than the 103 nat GILBO. This further reinforces the utility of the GILBO for cross-model comparisons.

## 4.3   Precision of the GILBO

While comparisons between models seem well-motivated, the SBC results in Section 4.2 highlight some mismatch in the variational approximation. How well can we trust the absolute numbers computed by the GILBO? While they are guaranteed to be valid lower bounds, how tight are those bounds?

To answer these questions, note that the GILBO is a valid lower bound even if we learn separate per-instance variational encoders. Here we replicate the results of Lipton & Tripathi (2017) and attempt to learn the precise $z$ that gave rise to an image by minimizing the $L^2$ distance between the produced image and the target ($|x - g(z)|^2$). We can then define a distribution centered on $z$ and adjust the magnitude of the variance to get the best GILBO possible. In other words, by minimizing the $L^2$ distance between an image $x$ sampled from the generative model and some *other* $x'$ sampled from the same model, we can directly recover some $z'$ equivalent to the $z$ that generated $x$. We can then do a simple optimization to find the variance that maximizes the GILBO, allowing us to compute a very tight GILBO in a very computationally-expensive manner.

Doing this procedure on the same three models as in Figures 6 and 7 gives (87, 111, 155) nats respectfully for the (41, 70, 104) GILBO models, when trained for 150k steps to minimize the $L^2$ distance. These approximations are also valid lower bounds, and demonstrate that our amortized GILBO calculations above might be off by as much as a factor of 2 in their values from the true generative information, but again highlights that the comparisons between different models appear to be real. Also note that these per-image bounds are finite. We discuss the finiteness of the generative information in more detail in Section 4.6.

Naturally, learning a single parametric amortized variational encoder is much less computationally expensive than doing an independent optimization for each image, and still seems to allow for comparative measurements. However, we caution against directly comparing GILBO scores from different encoder architectures or optimization procedures. Fair comparison between models requires holding the encoder architecture and training procedure fixed.

## 4.4 Consistency

The GILBO offers a signal distinct from data-based metrics like FID. In Figure 8, we visually demonstrate the nature of the retained information for the same three models as above in Figures 6 and 7. All three checkpoints for CelebA have the same FID score of 49, making them each competitive amongst the GANs studied; however, they have GILBO values that span a range of 63 nats (91 bits), which indicates a massive difference in model complexity. In each figure, the left-most column shows a set of independent generated samples from the GAN. Each of these generated images are then sent through the variational encoder $e(z|x)$ from which 15 independent samples of the corresponding $z$ are drawn. These latent codes are then sent back through the GAN's generator to form the remaining 15 columns.

The images in Figure 8 show the type of information that is retained in the mapping from image to latent and back to image space. On the right in Figure 8c with a GILBO of 104 nats, practically all of the human-perceptible information is retained by doing this cycle. In contrast, on the left in Figure 8a with a GILBO of only 41 nats, there is a good degree of variation in the synthesized images, although they generally retain the overall gross attributes of the faces. In the middle, at 70 nats, the variation in the synthesized images is small, but noticeable, such as the sunglasses that appear and disappear 6 rows from the top.

## 4.5 Overfitting of the GILBO Encoder

Since the GILBO is trained on generated samples, the dataset is limited only by the number of unique samples the generative model can produce. Consequently, it should not be possible for the encoder, $e(z|x)$, to overfit to the training data. Regardless, when we actually evaluate the GILBO, it is always on newly generated data.

Likewise, given that the GILBO is trained on the "true" generative model $p(z)p(x|z)$, we do not expect regularization to be necessary. The encoders we trained are unregularized. However, we note that any regularization procedure on the encoder could be thought of as a modification of the variational family used in the variational approximation.

The same argument is true about architectural choices. We used a convolutional encoder, as we expect it to be a good match with the deconvolutional generative models under study, but the GILBO would still be valid if we used an MLP or any other architecture. The computed GILBO may be more or less tight depending on such choices, though – the architectural choices for the encoder are a form of

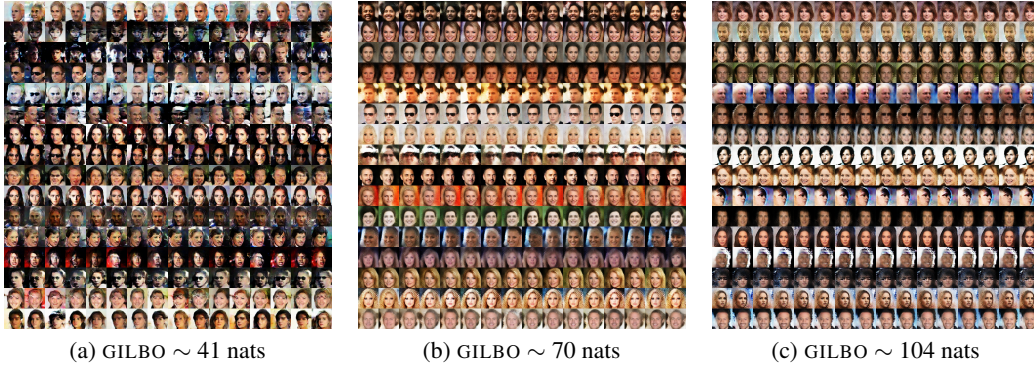

(a) GILBO $\sim$ 41 nats　　　　　(b) GILBO $\sim$ 70 nats　　　　　(c) GILBO $\sim$ 104 nats

Figure 8: Visual demonstration of consistency. The left-most column of each image shows a sampled image from the GAN. The next 15 columns show images generated from 15 independent samples of the latent code suggested for the left-most image by the trained encoder used in estimating the GILBO. All three of these GANs had an FID of 49 on CelebA, but have qualitatively different behavior.

## 4.6　Finiteness of the Generative Information

The generative mutual information is only infinite if the generator network is not only deterministic, but is also invertible. Deterministic many-to-one functions can have finite mutual informations between their inputs and outputs. Take for instance the following: $p(z) = \mathcal{U}[-1, 1]$, the prior being uniform from -1 to 1, and a *generator* $x = G(z) = \text{sign}(z)$ being the sign function (which is $C^{\infty}$ almost everywhere), for which $p(x|z) = \delta(x - \text{sign}(z))$ the conditional distribution of $x$ given $z$ is the delta function concentrated on the sign of $z$.

$$p(x, z) = p(x|z)p(z) = \frac{1}{2}\delta(x - \text{sign}(z)) \qquad p(z) = \int_{-1}^{1} dx\, p(x, z) = \frac{1}{2}\delta(z-1) + \frac{1}{2}\delta(z+1) \quad (6)$$

$$I(X; Z) = \int dx\, dz\, p(x, z) \log \frac{p(x, z)}{p(x)p(z)} \tag{7}$$

$$= \int_{-1}^{1} dx \int_{-1}^{1} dz\, \frac{1}{2}\delta(x - \text{sign}(z)) \log \frac{\delta(x - \text{sign}(z))}{\frac{1}{2}\delta(z - 1) + \frac{1}{2}\delta(z + 1)} \tag{8}$$

$$= \left[\frac{1}{2}\log 2\right]_{z=-1} + \left[\frac{1}{2}\log 2\right]_{z=1} = \log 2 = 1\text{bit} \tag{9}$$

In other words, the deterministic function $x = \text{sign}(z)$ induces a mutual information of 1 bit between $X$ and $Z$. This makes sense when interpreting the mutual information as the reduction in the number of yes-no questions needed to specify the value. It takes an infinite number of yes-no questions to precisely determine a real number in the range $[-1, 1]$, but if we observe the sign of the value, it takes one fewer question (while still being infinite) to determine.

Even if we take $Z$ to be a continuous real-valued random variable on the range $[-1, 1]$, if we consider a function $x = \text{float}(z)$ which casts that number to a float, for a 32-bit float on the range $[-1, 1]$ the mutual information that results is $I(X; Z) = 26$ bits (we verified this numerically). In any chain $Z \rightarrow \text{float}(Z) \rightarrow X$ by the data processing inequality, the mutual information $I(X; Z)$ is limited by $I(Z; \text{float}(Z)) = 26$ bits (per dimension). Given that we train neural networks with limited precision arithmetic, this ensures that there is always some finite mutual information in the representations, since our random variables are actually discrete, albeit discretized on a very fine grid.

## 5 Conclusion

We've defined a new metric for evaluating generative models, the GILBO, and measured its value on over 3200 models. We've investigated and discussed strengths and potential limitations of the metric. We've observed that GILBO gives us different information than is currently available in sample-quality based metrics like FID, both signifying a qualitative difference in the performance of most GANs on MNIST versus richer datasets, as well as being able to distinguish between GANs with qualitatively different latent representations even if they have the same FID score.

On simple datasets, in an information-theoretic sense we cannot distinguish what GANs with the best FIDs are doing from models that are limited to making some local deformations of the training set. On more complicated datasets, GANs show a wider array of complexities in their trained generative models. These complexities cannot be discerned by existing sample-quality based metrics, but would have important implications for any use of the trained generative models for auxiliary tasks, such as compression or representation learning.

A truly invertible continuous map from the latent space to the image space would have a divergent mutual information. Since GANs are implemented as a feed forward neural network, the fact that we can measure finite and distinct values for the GILBO for different architectures suggest not only are they fundamentally not perfectly invertible, but the degree of invertibility is an interesting signal of the complexity of the learned generative model. Given that GANs are implemented as deterministic feed forward maps, they naturally want to live at high generative mutual information.

Humans seem to extract only roughly a dozen bits ($\sim$ 8 nats) from natural images into long term memory (Landauer, 1986). This calls into question the utility of the usual qualitative visual comparisons of highly complex generative models. We might also be interested in trying to train models that learn much more compressed representations. VAEs can naturally target a wide range of mutual informations (Alemi et al., 2017). GANs are harder to steer. One approach to make GANs steerable is to modify the GAN objective and specifically designate a subset of the full latent space as the informative subspace, as in Chen et al. (2016), where the maximum complexity can be controlled for by limiting the dimensionality of a discrete categorical latent. The remaining stochasticity in the latent can be used for novelty in the conditional generations. Alternatively one could imagine adding the GILBO as an auxiliary objective to ordinary GAN training, though as a lower bound, it may not prove useful for helping to keep the generative information low. Regardless, we believe it is important to consider the complexity in information-theoretic terms of the generative models we train, and the GILBO offers a relatively cheap comparative measure.

We believe using GILBO for further comparisons across architectures, datasets, and GAN and VAE variants will illuminate the strengths and weaknesses of each. The GILBO should be measured and reported when evaluating any latent variable model. To that end, our implementation is available at `https://github.com/google/compare_gan`.

### Acknowledgements

We would like to thank Mario Lucic, Karol Kurach, and Marcin Michalski for the use of their 3200 previously-trained GANs and VAEs and their codebase (described in Lucic et al. (2017)), without which this paper would have had much weaker experiments, as well as for their help adding our GILBO code to their public repository. We would also like to thank our anonymous reviewers for substantial helpful feedback.

## Footnotes

[2]Note that a *new* $e(z|x)$ is trained for both GANs and VAEs. VAEs do not use their own $e(z|x)$, which would also give a valid lower bound. In this work, we train a new $e(z|x)$ for both to treat both model classes uniformly. We don't know if using a new $e(z|x)$ or the original would tend to result in a tighter bound.

[3]Note that Wu et al. (2017) is complementary to our work, providing both upper and lower bounds on the log-likelihood. It is our opinion that their estimates should also become standard practice when measuring GANs and VAEs.

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
