[Reviews · NeurIPS 2018]

Reviewer 1



Overall I think is a very good paper and it is one of the better papers I've seen addressing evaluating GANs. I myself are fairly skeptical of FID and have seen other works criticizing that approach, and this work sheds some light on the situation. I think anyone who follows this work would be better informed than work that introduced inception or FID in how to evaluate GANs. That said, there is some missing discussion or comparison to related work (notably mutual information neural estimation (MINE) by Belghazi et al, 2018) as well as some discussion related to the inductive bias and boundedness of their estimator. I'd like to see a discussion of these things. Comments: Don’t forget "Quantitatively Evaluating GANs With Divergences Proposed for Training” (Im et al, ICLR 2018) as another method for evaluating GANs. Since you are estimating mutual information, it would be worth comparing to MINE (Belghazi et al, ICML 2018), or at least a good discussion relating to this model. Is there any way to tell if your encoder is overfitting? What would be the consequence of overfitting in your model? For instance, could your encoder preference to learn to map trivial noisy information from x to z, ignoring or overlooking important but more difficult to encode structure? Your generator and encoder are deterministic and the variables continuous, are there issues with the boundedness of the mutual information? If so, how well do we expect this estimator to do as the true mutual information between the generator input and output is high? In addition, how can regularization of your encoder (a la Roth et al ICML, 2017 or Mescheder et al ICML 2018) change the quality of the estimator (is it better or worse)? Furthermore, what is the potential effect of the inductive bias from the choice of convolutional neural network as your encoder? -------------- I have read the response, and I am happy with the response. However, I encourage the authors to think more about the potential unboundedness of the KL mutual information and the boundedness of their family of functions / regularization.

Reviewer 2



Summary ======= This paper proposes a metric to evaluate generative models. It suggests to use a variational lower bound on the mutual information between the latent variables and the observed variables under the distribution of the model. Good ==== The most interesting finding of this paper is that the proposed metric sits close to ln(training_set_size) nats for models with low FID on MNIST and FashionMNIST (although not for CelebA or CIFAR), which the authors suggest might be because the models memorized the training set. If the authors understood this result better and provided additional evidence that GANs are memorizing the training set, it might make for an interesting paper. Bad === The proposed metric is in general not well defined for GANs mapping z to x via a deterministic function. The mutual information is expressed in terms of a differential entropy using densities, but those densities might not exist. Even if it was properly defined, the mutual information is likely to diverge to infinity for GANs. I suspect that the estimated finite values are an artefact of the approximation error to the posterior p(z | x), and it is therefore not clear how to interpret these values. The authors claim "any failure to converge for the approximate encoder to match the true distribution does not invalidate the bound, it simply makes the bound looser." Yet if the mutual information is infinite, the bound is not just loose but may be completely meaningless, which seems like a big problem that should be addressed in the paper. The authors suggest that the proposed metric can be used as a measure of "complexity of the generative model". Yet simple linear models (PCA) would have infinite mutual information. I would be inclined to change my score if the authors can provide an example of a deterministic differentiable functios g(z) where I(z, g(z)) isn't either 0 or infinity. Of course, adding a bit of noise would be one way to limit the mutual information, but then I'd argue we need to understand the dependence on the noise and a principled way of choosing the noise level before the metric becomes useful. At a higher level, the authors seem to (incorrectly) assume that the goal of generative modeling is to generate visually pleasing images. Yet this is probably the least interesting application of generative models. If generating realistic images was the goal, a large database of images would do such a good job that it would be hard to justify any work on generative models. For more well-defined tasks, measuring generalization performance is usually not an issue (e.g., evaluating unsupervisedly learned representations in classification tasks), diminishing the value of the proposed metric.

Reviewer 3



[Response to rebuttal: I am happy and thoroughly satisfied with the author's response to the reviews, which is why I have increased my rating. I only want to touch upon one aspect once more: Reviewer 2 has raised an interesting issue (and in my view the authors have responded well and convincingly), namely the problem that the continuous MI might diverge for a differentiable deterministic model. The paper might benefit from explicitly including and addressing this issue, at least in the appendix. The issue is trivially resolved by acknowledging the fact that we perform computation with limited bit-precision, so the authors might even choose to make this explicit by replacing integrals with sums in their equations. For the continuous case, the problem remains when using a deterministic, reversible generator and I think it is interesting to comment on this in the paper and discuss how (and potentially under what circumstances) typical Relu networks tend to produce non-reversible functions ("many-to-one" mapping) while still being (mostly) differentiable. A full investigation of this is beyond the scope of the paper, but it might spark interest to look into this in some future work and thoroughly investigate what kind of compression GAN decoders perform (in the continuous limit).] The paper introduces a new metric to quantitatively characterize the complexity of a generative latent-variable model with the ultimate goal of providing a performance indicator to quantitatively characterize generative models obtained from both GANs and VAEs. The aim is to quantify the mutual information between the latent variable z and the generated data x. This quantity is interesting since it provides a training-data independent measure, is invariant under re-parametrization and has interesting and well-known information-theoretic properties and interpretations. Since the computation of the mutual information involves the intractable posterior p(x|z), the paper proposes to instead consider a lower bound (GILBO) based on a variational approximation of the posterior instead. The measure is demonstrated in an exhaustive experimental study, where 800 models (VAEs and GANs) are compared on four different datasets. The authors find that the proposed measure provides important insight into the complexity of the generative model which is not captured by previously proposed data-dependent and sample-quality focused measures such as the FID score. The GILBO allows to qualitatively divide trained models into different regimes such as “data memorization” or “failure to capture all classes in latent space”. Additionally the GILBO highlights some qualitative differences between GANs and VAEs. The paper concludes by addressing some potential shortcomings of the method: reproducibility of quantitative values, tightness of lower bound, quality of variational approximation and consistency with qualitative “behavior” of the generative model. The recent literature reports several attempts at objectively comparing the quality of complex generative models obtained from GANs or VAEs. While many methods so far focused on capturing the match between the true and the generated data-distribution or quantitative measures that correlate well with visual quality of generated samples, these measures seem to be insufficient to fully capture all important aspects of a generative model. Reporting the GILBO as an additional quantity that characterizes complexity of the generative model is a novel and original proposal. The paper is generally well written (except for some hiccups at the beginning of section 2) and the experimental section is impressive (certainly in terms of invested computation). Main points of criticism regarding the method are already addressed in the paper. For these reasons, I argue for accepting the paper. My comments below are meant as suggestions to authors for further increasing the quality of the paper. (1 Intro of the GILBO in Sec. 2) Section 2 (lines 23 to 48) could use some polishing and additional details, particularly for readers unfamiliar with standard notation in generative latent-variable models. In particular: -) please explain what the variables X and Z correspond to in the context of generative models, ideally even mention that such a generative model consists of a prior over the latent variable and a (deterministic) generator p(x|z) which is a deep neural network in GAN and VAE. -) line(27): I guess to be precise, it is the reduction in the number of questions to guess X if you observe Z and know p(X). -) line (32): I think it would not hurt if you (in-line) add the equation for marginalization to make it obvious why p(x) is intractable. Also consider giving the equation for the posterior p(z|x) as you have it in line 34 before mentioning p(x), since p(x) does not appear in Eq. (1) but p(z|x) does. This might help readers follow the reasoning more easily. -) Eq (2): perhaps consider being explicit about the fact that GILBO is a function which depends on the parameters of e(z|x), for instance by writing GILBO(\theta)=… and then using e_\theta(z|x) or similar. -) line (42 – 44): “maximizing the likelihood…” – it could be helpful to write down the same as an equation, just to make it precise for readers. (2 Proofs) There are two arguments that you might want to back up with a proof (e.g. in the supplementary material). Though the proofs should not be too difficult, some readers might appreciate them and they are actually central statements for the appealing properties of the GILBO. Additionally it might be beneficial because it would require being more specific about the technical conditions that need to be fulfilled to get the appealing properties of GILBO. -) line 46: show that the GILBO is actually a lower bound to the mutual information. -)line 48: show that failure to converge, etc. simply makes the bound looser, i.e. show that Eq. (2) cannot diverge due to some ill-parameterized e(z|x). (3 Discussion) Three more issues that you might want to include in your discussion. -) Do you think it would be possible/computationally feasible to replace the variational approximation of the posterior with some sampling-approximation, which would in principle allow for arbitrary precision and thus guarantee convergence to the actual mutual information (while computationally expensive, this could be used to compute a gold-standard compared to the variational approximation)? -) How sensitive is the GILBO to different settings of the optimization process (optimizer, hyper-parameters)? Ideally, this answer is backed up by some simulations. This would be of practical importance, since comparing GILBOs of different models could in principle require following very closely the same procedure (and of course encoder architecture) across papers. -) Besides qualitative “sanity-checks” (as in Fig. 2: which regime does my generative model fall into), how do you propose to use the GILBO to quantitatively compare models. Say I have two models with different GILBO (but in the same qualitative regime) and different FID score – which model is favorable, higher GILBO, higher FID, …? Misc: -) Footnote 1, last sentence: … it could also achieve a looser bound -) Figure 1a: Any specific reason for a different color-map here, otherwise please use the same color-map as in the other plots. -) Fig. 7 caption vs. line 133: did you use 128 or 127 samples? -) Fig. 8 caption vs. line 165/166: did you use 16 or 15 samples? -) citations: I quite like the author, year citations (even though they “cost” extra space in a conference paper) – please check that all your citations have the year properly set (see line 194, where it is missing) -) line 207/208: Reporting the GILBO as an additional measure could be a good idea, please be a bit more specific that this would require fixing the encoder architecture and optimization procedure as well and comment on practical aspects such as whether you would use a different encoder architecture per data-set or the same across many data-sets, etc.